# An Overview of Nanoparticle-Based Delivery Platforms for mRNA Vaccines for Treating Cancer

**DOI:** 10.3390/vaccines12070727

**Published:** 2024-06-29

**Authors:** Yang Lin, Xuehua Chen, Ke Wang, Li Liang, Hongxia Zhang

**Affiliations:** 1Department of Pathology, School of Basic Medical Sciences, Southern Medical University, Guangzhou 510515, China; liny991223@163.com (Y.L.); ansonchan0320@163.com (X.C.); qy19806583152@163.com (K.W.); 2Guangdong Provincial Key Laboratory of Molecular Tumor Pathology, Guangzhou 510515, China; 3Department of Pathology, Nanfang Hospital, Southern Medical University, Guangzhou 510515, China; 4Jinfeng Laboratory, Chongqing Science and Technology Innovation Center, Chongqing 401329, China

**Keywords:** vaccine platform, nanoparticles, mRNA vaccine, lipid-based nanoparticles, anti-tumor immunity

## Abstract

With its unique properties and potential applications, nanoparticle-based delivery platforms for messenger RNA (mRNA) vaccines have gained significant attention in recent years. Nanoparticles have the advantages of enhancing immunogenicity, targeting delivery, and improving stability, providing a new solution for drug and vaccine delivery. In some clinical studies, a variety of nanoparticle delivery platforms have been gradually applied to a wide range of vaccine applications. Current research priorities are exploring various types of nanoparticles as vaccine delivery systems to enhance vaccine stability and immunogenicity. Lipid nanoparticles (LNPs) have shown promising potential in preclinical and clinical studies on the efficient delivery of antigens to immune cells. Moreover, lipid nanoparticles and other nanoparticles for nucleic acids, especially for mRNA delivery systems, have shown vast potential for vaccine development. In this review, we present various vaccine platforms with an emphasis on nanoparticles as mRNA vaccine delivery vehicles. We describe several novel nanoparticle delivery platforms for mRNA vaccines, such as lipid-, polymer-, and protein-based nanoparticles. In addition, we provide an overview of the anti-tumor immunity of nanovaccines against different tumors in cancer immunotherapy. Finally, we outline future perspectives and remaining challenges for this promising technology of nanoparticle-based delivery platforms for vaccines.

## 1. Introduction

As a means of preventing infectious diseases, vaccination has been demonstrated to be the most efficient and cost-effective. Vaccine administration can activate the immune system and stimulate innate and adaptive immunity against pathogens. In recent years, the use of nanoparticles in vaccine platform development has garnered considerable attention due to their unique properties and potential applications. The field of disease treatment has improved with the development of smart nanocarriers, which have the ability to transport hydrophilic and hydrophobic medications under controlled conditions [1,2]. Cancer nanovaccines represent an innovative therapeutic approach that can co-deliver immune adjuvants and tumor antigens. Currently, there are approximately 50 cancer nanomedicines that have obtained United States Food and Drug Administration (FDA) approval, underscoring the predominant focus of nanomedicine on cancer treatment [3]. Nanoparticles, as a novel drug delivery system, have demonstrated extensive prospects in the field of medicine. The advantages of nanoparticles as vaccine delivery vehicles include their ability to enhance immunogenicity, target delivery, and improve stability. Vaccine delivery platforms are used to deliver antigens and adjuvants to the immune system that can enhance the stability and immunogenicity of vaccines. Nowadays, nanoparticles serving as delivery platforms for vaccines, especially messenger RNA (mRNA) vaccines, are being actively investigated. There are various types of nanoparticles being researched, including lipid-based, peptide-based, polyplexed and polymeric nanoparticles, and inorganic nanoparticles [4]. 

In the past few years, mRNA vaccine technology has drawn increasing research and commercial attention owing to their potential in tackling both infectious disease prevention and cancer therapeutics [5,6,7,8]. mRNA vaccine administration delivers the mRNA to host cells; then, the cells take up the mRNA and translate it into the encoded protein. For example, in the nanoparticle delivery platform, lipid-based nanoparticles are designed to encapsulate mRNAs and to deliver mRNAs encoding antigens to immune cells. In particular, mRNA vaccines have been urgently approved for use in the coronavirus disease 2019 (COVID-19) pandemic. To overcome the barriers to safe and effective RNA delivery, scientists have developed various delivery systems that protect the mRNA from degradation, thus maximizing its delivery to target cells. The advances in synthetic materials as delivery vehicles that encapsulate mRNA, such as lipid nanoparticles and polymers, have focused on non-viral-based delivery systems [9]. Pfizer/BioNTech’s Comirnaty (NT162b2) and Moderna’s Spikevax (mRNA-1273) were the first mRNA products to obtain approval from the FDA or the European Medicines Agency (EMA) and have been broadly validated in clinical applications [10,11]. This indicates the therapeutic potency of lipid-nanoparticle-based mRNA vaccines for tumor treatment. Besides lipid-nanoparticle-based delivery platforms for vaccines, the development of other nanoparticles has also gained tremendous momentum. For instance, the use of inorganic nanoparticles such as gold, silver, and iron oxide nanoparticles for vaccine delivery is also a focus of ongoing research [12].

Furthermore, researchers have also been exploring the potential of nanoparticles to deliver antigens to specific immune cells such as dendritic cells (DCs) or tissues [13]. Generally, the surface modification of nanoparticles with targeting ligands that can selectively bind to receptors on immune cells enhances the uptake of vaccines and the activation of immune responses. In addition to vaccine delivery, current research studies are also focused on understanding the immunological mechanisms underlying the interaction between nanoparticles and the immune system. This includes investigating the impact of nanoparticle size, shape, surface charge, and surface chemistry on immune cell activation, antigen presentation, and the generation of immune responses [14]. Overall, the development of nanoparticle-based delivery platforms is characterized by a diverse range of studies aimed at optimizing their design, formulation, antigen selection, and so on. Here, we summarize the advances in nanoparticle-based delivery platforms for vaccine development, especially for mRNA vaccines, and their potential prospects in clinical applications.

## 2. Cancer Vaccines 

### 2.1. Traditional Tumor Vaccine Types

Tumor vaccines immunize the body using tumor-specific antigens to stimulate the process of antigen presentation, enhancing the immune system’s recognition of and attack on tumors. The types of tumor vaccines mainly include tumor lysate vaccines, dendritic-cell-based vaccines, peptide vaccines, nucleic acid (such as mRNA) vaccines and tumor neoantigen vaccines. Among them, dendritic cell vaccines are the most commonly studied cell-based tumor vaccines [15]. Dendritic-cell-based vaccines are DCs from patients loaded with or expressing tumor-associated antigens (TAAs), which can trigger the activation of T cells to attack tumor cells.

A dendritic-cell-based nanovaccine that regulates lipid metabolism and stimulates the innate immune response (TOFA@PLGA@OMs-PLs, named as TPOP) was developed by Qin et al. to enhance and restore the cross-presentation ability of DCs through the regulation of lipid metabolism [16]. This vaccine is administered following tumor cell death induced by doxorubicin (DOX) in order to capture the TAAs. TPOP could inhibit lipid accumulation and restore the progress of DCs’ cross-presentation. Peptide vaccines are a type of vaccine that utilize tumor-specific peptide fragments as antigens. These peptide fragments are typically selected from tumor-associated proteins and can be recognized by the immune system [17]. Tumor neoantigen vaccines are specifically designed vaccines that target new antigens presented on tumor cells. These neoantigen vaccines are an effective means of stimulating, enhancing, and inducing anti-tumor T cell responses with good feasibility, a high level of safety, and simple preparation. Overall, with the injection of tumor vaccines, T cells can be activated to lead tumor elimination. Various forms of vaccines, including neoantigen-based dendritic cells, nucleic acids, peptides, and multi-epitope neoantigens, are being evaluated through clinical trials in a variety of tumors. 

Traditional cancer vaccines are designed to activate the immune system, enabling it to identify and attack cancer cells that possess these same antigens. Nevertheless, the complex and diverse characteristics of tumors pose significant challenges of limited efficacy and lack of specificity [18]. The limitations of conventional vaccines may impose constraints on their utilization in disease prevention and treatment [19]. For example, the patient’s own cells are needed for the preparation of dendritic cell vaccines. Due to MHC restriction, peptide vaccines are selectively activated monoclonal T cells and thus pose a significant risk of immune evasion [20]. 

### 2.2. mRNA Vaccines

In this decade, nucleic acid vaccines have emerged as innovative vaccines. Nucleic acid vaccines, such as mRNA vaccines, serve as a promising new form of vaccines that relies on using nucleic acids as information carriers. Notably, DNA- or RNA-based vaccines have gradually evolved into a promising form of conventional vaccines [21]. These vaccines display their preventive and treatment role by delivering exogenous nucleic acids into immune cells. Nucleic acid vaccines transmit the coding information of nucleic acids to the cells, eliciting the production of the encoded antigen. These antigens can be recognized by the immune system, processed, and presented to T cells. With the activation of T cells, the vaccine can trigger an immune response against tumor-specific antigens and initiate an attack on tumor cells. Compared with mRNA vaccines, DNA vaccines carry the risk of genomic alterations, long-term expression, and the generation of anti-DNA autoantibodies that might impede their utilization in humans [22,23].

The field of tumor vaccines has seen multiple clinical trials for mRNA vaccines, particularly before the COVID-19 pandemic, which distinguish them from DNA vaccines [24,25]. mRNA vaccines, particularly those based on individual-specific neoantigens, represent a new direction in cancer therapy. In comparison with other nucleic acid vaccines, the advantages of mRNA-based cancer vaccine strategy include the following: (1) mRNA can be translated into protein rapidly in both non-dividing and hard-to-transfect cells such as DCs without the need for nuclear translation and transcription. The rate and magnitude of protein expression from mRNA are typically higher than from DNA vaccines. (2) Unlike DNA vaccines, mRNA vaccines will not influence genetic material or integrate into the genome sequence; thus, they are free from the risk of insertional mutagenesis [26,27]. (3) Other advantages of mRNA vaccines include the high potential for their rapid development and potency, low-cost manufacture, and safe administration [18,19]. Now, mRNA vaccines have shown promise in the development of personalized cancer immunotherapy.

## 3. Nanotechnology in mRNA Vaccine Platforms

The development of mRNA technology, especially its successful application in the development of COVID-19 vaccines, has also greatly propelled advancements in the field of cancer vaccines. mRNA vaccines can be rapidly designed and produced, effectively encoding tumor-specific antigens to guide the immune system in recognizing and eliminating cancer cells. This innovative technology provides a more flexible and effective approach for the development of cancer vaccines [18]. In this review, we mainly describe nanoparticle delivery platforms for mRNA vaccines, such as lipids, polymers, proteins, virus-like particles, and cationic nanoemulsion-based nanoparticles (Figure 1). Effective in vivo mRNA delivery is crucial for achieving the desired therapeutic effects. Nevertheless, due to the intrinsic structural features of mRNA, such as its substantial molecular weight, potent negative charge, and susceptibility to degradation [28], the distribution effectiveness of mRNA in vivo is limited, and its stability remains a significant challenge for the development and utilization of mRNA vaccines.

Therefore, the efficient delivery into cells of antigen-encoding mRNA that has been transcribed in vitro is a key challenge in mRNA cancer vaccine applications. With the advancement of nanomedicine, the application of nanomaterials in tumor immunotherapy is becoming increasingly widespread. As antigens and adjuvants need to be co-delivered into antigen-presenting cells for effective antigen presentation, biomimetic nanoparticles with high drug-loading capacity and good drug-release controllability are ideal carriers for vaccines [13]. Their degradable shell can help in the sustained release of drugs upon reaching the designated site [29]. Nanoparticles are classified as a material or substance with a dimension between 1 to 100 nm or up to 1000 nm. As a delivery vehicle, nanoparticles are usually equipped with three major roles: as carriers, adjuvants, and presentation platforms [30]. Here, we will describe several common nanoparticle delivery platforms and their current research status.

### 3.1. Lipid-Based Nanoparticle Delivery Systems

#### 3.1.1. Composition of Lipid-Based Nanoparticles 

The FDA has granted approval for the use of lipid-based nanoparticles (LNPs) as a crucial drug delivery system specifically for delivering siRNA and mRNA [31,32]. The structure of lipids consists of three distinct domains: a polar head group, a hydrophobic tail region, and a linker connecting the two domains [29,33]. To prevent ribonuclease attacks and to promote endo-lysosomal escape, mRNA has been packaged into nanoparticles as part of their core or adsorbed onto the lipid bilayers [34]. Lipids have been developed for mRNA delivery, include cationic lipids, ionizable lipids, and other types of lipids, such as phospholipids, cholesterol or polyethylene glycol (PEG) lipids (Figure 2) [35]. The ability of lipid components to self-assemble into nanostructured entities is crucial for the formation of lipoprotein nanoparticles (LNPs). An electrostatic link is initially formed between positively charged lipids and negatively charged nucleic acids, followed by the aggregation of lipid components through van der Waals and hydrophobic interactions. There are various methods available for preparing LNPs, including microfluidic techniques, film hydration, liposome extrusion, and nano-precipitation. Among these methods, quick mixing of lipid and aqueous phase components is commonly employed. The microfluidic approach has become the preferred method for preclinical research due to its high reproducibility [36].

#### 3.1.2. Stability and Modification of Lipid-Based Nanoparticle Delivery Platforms

Stability is crucial for LNP formulation in industrial production. The stability of nanoparticles is primarily influenced by the particle size distribution, steric hindrance, and electrostatic repulsion. The particle size distribution can determine their administration, shelf-life, and biodistribution [37]. Steric hindrance involves the introduction of macromolecules, polymers, or other hydrophobic components onto the surface of nanoparticles to hinder their proximity and aggregation. Electrostatic stabilization is achieved by regulating the surface charge of nanoparticles, ensuring that all particles in the solution carry the same charge. The resulting electrostatic repulsion between the charges effectively prevents particle aggregation. This process typically involves modifying molecules or ions so that they possess charged surfaces [38]. Cationic surfaces can enhance the electrostatic repulsion between particles, thereby reducing the likelihood of aggregation and precipitation. Similarly, surface modifications with negative charges can also amplify the electrostatic repulsion, thus enhancing particle dispersion and stability. Moreover, PEG modification can augment the hydrophilicity of LNPs in aqueous solutions, diminish the hydrophobic interaction among particles, and facilitate their attachment to nanoparticle surfaces through physical adsorption or chemical bonding, ultimately improving their dispersion stability [39]. By modifying the functional groups or adjusting the solution conditions of the particle surface, it is possible to flexibly manipulate the charge characteristics of the particle surface. The double stabilization effect can be achieved through the introduction of molecules on the particle surface that offer both steric hindrance and electrical energy, such as through the surface modification of charged polymers or biomacromolecules. Usually, LNPs can effectively protect the mRNA from degradation and damage while maintaining good stability. Those characteristics are crucial for a delivery of vaccines that can help to avoid immune and toxic reactions [40]. However, the stability of LNPs can be influenced by lipid polymorphisms, sterilization, and phase separation [41]. For example, as a current sterilization technique, γ-irradiation could induce chemical degradation of the lipids [42]. Sterile filtration can be applied to particles with a size lower than the filter pores. Therefore, suitable techniques should be selected in lipid nanoparticle production.

LNP-based delivery platforms are safe and suitable for nucleic acid and therapeutic cargo delivery. Now, it is commonly used in vaccine delivery and development. Due to the similarity of the lipid components to those found in the body, LNPs exhibit excellent biocompatibility. Moreover, the surface of LNPs can be modified. Surface modifications can usually improve the targeting property of nanoparticles. Typically, they are prepared by conjugating a small molecule ligand, peptide, or monoclonal antibody to the surface of the LNP in order to achieve surface attachment of the ligand for the specific recognition and binding to a receptor on the cell [43]. The surface modification strategies of LNPs can be categorized into “pre-modification” and “post-modification”. The “pre-modification” approach involves either pre-conjugating ligand molecules to lipid molecules or incorporating ligand lipids onto the surface of LNPs, facilitating self-assembly with other components to produce LNPs. On the other hand, in the “post-modification” technique, ligand molecules are directly conjugated to the surface of LNPs through affinity adsorption or chemical interactions [44]. The advantage of post-modification lies in its exceptional flexibility, which allows for the specific conjugation of antibodies or antibody fragments onto the surface of LNPs, thereby enabling precise targeting toward particular cells or tissues. This feature allows for the precise delivery of antigens to specific cells or tissues, thereby enhancing the treatment efficacy of the vaccine [36]. Post-modification not only imparts novel functionalities to LNPs but also exerts a substantial influence on their physical, chemical, and biological stability. However, certain post-modification methods may result in compromised binding affinity of the functional molecules, thereby potentially impacting the long-term stability of LNPs. When deciding between pre-modification and post-modification approaches, a comprehensive evaluation should be undertaken. If a highly flexible and versatile LNP is desired, post-modification may be deemed more suitable. Conversely, if enhanced stability and uniformity are pursued, pre-modification may offer better outcomes.

Another key characteristic to be considered when designing LNPs for drug delivery, especially nucleic acid therapy, is the ionizable lipid’s ionization state in the system, which is characterized by its apparent pKa [45]. The degree of ionization significantly influences the surface charge, stability, toxicity, and in vivo efficacy of nanoparticles. Many factors can determine the apparent pKa of ionizable lipids, such as the lipid structure, various components in the LNPs’ formulation, the nanoparticle size, and the pH of the bulk solution [46]. Previous studies have shown that for effective nucleic acid delivery, the optimal pKa of LNPs should be around pH 6.5. For example, lipid-based nanoparticles have the most important role and were the first type of vector developed for mRNA delivery. They contain amine cations and carry positive charges that can achieve adsorption and complexation of negative mRNA through electrostatic interaction. However, this positivity is potentially cytotoxic. The appearance of ionizable lipids makes up for the deficiency of cationic lipids. Ionizable liposomes are negatively charged at acidic pHs and electrically neutral under physiological conditions in vivo, which avoid non-specific recruitment with serum proteins and reduce cytotoxicity. These properties are responsible for the high serum stability and long blood circulation time [47]. After endocytosis into endosomes, ionized lipids could reverse to being positive at low pH and promote the endosomal escape of mRNA by a “proton sponge” effect, thus improving the transfection efficiency [48]. 

In conclusion, a comprehensive understanding of the structural properties of different types of lipids in LNP systems and their pivotal role in facilitating nucleic acid delivery will contribute to the enhanced design of next-generation lipids.

#### 3.1.3. Application of Lipid-Based Nanoparticle Delivery Platforms

mRNA vaccines can enhance the immune response against tumors by stimulating and strengthening the cytotoxic function of CD8^+^ T cells, leading to increased tumor clearance rates and potentially preventing tumor recurrence. Effective delivery vehicles are essential to the severe acute respiratory syndrome coronavirus 2 (SARS-CoV-2) mRNA vaccines that have now been authorized as well as many mRNA-based preventive vaccines that are in clinical development. Serving as delivery vehicles, LNPs can effectively facilitate mRNA expression in situ and confer intrinsic adjuvant properties to the vaccine [49]. Comparative assessments of three mRNA-LNP vaccines (unmodified (U), m1Ψ-modified non-replicating (nr) mRNA, and self-amplifying (sa) mRNA) encoding the identical antigen, gDE7 (the fusion of the HPV-16 E7 oncoprotein and the herpes simplex virus type 1 glycoprotein D), have been documented in multiple studies investigating a preclinical model of HPV-16-associated malignancies [50]. Clearance rates of tumors are associated with the robust CD8^+^ T cell response induced by gDE7 mRNA-LNP vaccination, which also effectively prevents tumor recurrence. Additionally, it has been demonstrated that CD4^+^ T cells contribute to the establishment of long-term immunological memory and the efficient priming of CD8^+^ T cells in anti-tumor models [50]. To elicit a robust CD8^+^ T cell response, Chen et al. employed lipid-nanoparticle-mediated administration of an mRNA cancer vaccine targeted specifically to the lymph nodes [51]. By encapsulating mRNA in lipid nanoparticles, the immunogenicity and antigen presentation efficiency of the mRNA vaccine is enhanced, enabling its effective targeted delivery to lymph nodes, activating CD8^+^ T cells, and thereby enhancing the immune system’s antigen-specific response to the tumor. A recent study reported a novel nanosized lipid polymer system (PIR-Au NPs) based on enhanced penetration and retention (EPR) effects, which can be successfully applied to image-guided passive targeted therapy [52]. This vaccine may provide a promising tool for the future treatment of cancer. Zhang et al. developed a lipid-like material, C1, with a 12-carbon tail that could effectively deliver mRNA into DCs, produce efficient mRNA translation, and induce a robust T cell response [53]. Overall, these research studies highlight that mRNA vaccines can enhance the immune response against tumors by stimulating and strengthening the cytotoxic function of CD8^+^ T cells.

As a key component of the human immune system, T cells are the focus of drug delivery in many medical applications [54]. Transfecting T cells poses a particular challenge in comparison with other cell types. Engineered T cells expressing chimeric antigen receptors (CARs) were efficacious therapies against hematological malignancies. The direct transfer of in vitro transcribed CAR-mRNA by electroporation (EP) into T cells has been demonstrated as a promising strategy for CAR-T cell engineering [55]. However, its application in vivo is hindered by the potential induction of severe cytotoxicity and technical challenges. The clinical trials of CAR-T cells generated by EP showed poor anti-tumor responses. Currently, numerous studies have successfully generated CAR-T cells in vitro using mRNA-LNP platforms, demonstrating promising preclinical anti-tumor effects. LNPs provides a more streamlined approach for the production of mRNA-based CAR-T cells. The Wender and Waymouth team at Stanford University previously reported on a novel class of mRNA delivery systems known as CARTs [56]. The positive fragments of CARTs bind to mRNA to form nanoparticles in an acidic environment, but they are rapidly degraded and neutralized in the intracellular environment for efficient mRNA release. The CAR-T-based mRNA vaccine delivery system represents an emerging therapeutic strategy that combines the advantages of CAR-T and mRNA vaccine technology. These systems are specifically designed to employ targeted delivery mechanisms for directly administering mRNAs encoding CAR-T cells to patients, with the ultimate goal of generating therapeutic CAR-T cells in vivo.

The remarkable advancements of CAR-T cell immunotherapy are overshadowed by the challenges, which include tumor heterogeneity and the inhibitory microenvironment in the clinical transformation of solid tumors. Consequently, numerous researchers are actively striving to overcome these obstacles. Li et al. reported a new CAR-T system called beta-amido carbonate (bAC)-CARTs [57]. The bAC-CARTs showed superior T cell transfection efficiency both in vitro and in vivo. Zhang et al. developed a novel approach for the precise modulation of guanine ion activity through charge offset, leading to the development of an extrahepatic targeted, regulated, and predictable GSer-CART transporter [58]. This innovative strategy enabled the efficient in vivo delivery of modified mRNA, effectively triggering immune responses and suppressing tumor growth. Above all, although there are still important questions that need to be addressed regarding the novel mRNA CAR-T cells, we believe that both CAR-T and mRNA technologies hold promising potential for applications in cancer vaccine development to address unmet medical needs beyond the field of oncology.

In summary, LNP delivery platforms have shown promising clinical applications. However, further efforts are required to enhance the translation efficiency, stability, and safety of these approaches [59,60,61]. 

#### 3.1.4. Disadvantages and Optimization of Lipid-Based Nanoparticle Delivery Platforms

The LNP-based mRNA vaccines approved by the FDA for clinical are capable of protecting mRNAs against protease degradation and of facilitating their cellular uptake across the cell membrane. However, they still face several unresolved challenges. The administration of mRNA cancer vaccines typically involves repetitive injections, which can lead to lipid accumulation in tissues over time and clearance by anti-drug antibodies. Notably, the majority of LNPs accumulate in the liver, contradicting the initial objective of mRNA vaccines in achieving their intended function [58]. Researchers should endeavor to modify LNPs in order to manipulate organ targeting. Alongside direct modifications of LNPs themselves, local administration techniques such as intramural injection through lymph nodes can also enhance the targeting potential of LNPs. This approach prevents their systemic distribution to specific organs and contributes to the selective delivery of LNPs. 

After entering the target cells, LNPs must facilitate the mRNAs’ escape from the endosomal membrane for translation into functional proteins. However, most mRNAs are degraded in lysosomes following endocytosis, with only a small fraction successfully evading this fate and reaching the cytoplasmic compartment. Thus, there remains a need to improve delivery efficiency. To modify the surface characteristics of LNPs, Zhang et al. employed a straightforward fluorination technique involving the creation of fluorinated PEG–lipids through effective condensation reactions [62]. This alteration can improve mRNA translation efficacy by encouraging endosomal escape and cellular absorption. Moreover, the LNP-based mRNA vaccines have higher levels of reactive oxygen species formation during translation, which result in a reduction in translation efficiency as well as inflammation and other side effects. Yang et al. developed innovative LNPs (G-LNPs) for mRNA vaccines by synthesizing a lipid-modified poly(guanidine thioctic acid) polymer [63]. By eliminating reactive oxygen species that hinder translation and trigger inflammatory responses, G-LNPs significantly enhance the translational efficiency of loaded mRNAs while minimizing inflammation post-vaccination. A major hurdle in eliciting robust anti-tumor immune responses lies in the insufficient accumulation of LNP-based mRNA vaccines within antigen-presenting cells. Lei et al. developed DC-targeting LNPs using mannose-receptor-mediated endocytosis [64]. These sugar-coated LNPs (STLNPs-Man) could deliver mRNA to DCs effectively in vitro and in vivo. Importantly, STLNPs-Man@mRNA possessed the ability to downregulate cytotoxic T lymphocyte-associated antigen-4 (CTLA-4) expression by blocking the CD206/CD45 axis, showing enhanced anti-tumor efficacy through combined immune checkpoint blockade therapy.

The enhancement of mRNA-LNP specificity and the mitigation of off-target expression can enhance the efficacy of vaccine delivery systems. While the LNPs utilized in the COVID-19 vaccines were not specifically targeted and were primarily taken up by phagocytic cells (such as dendritic cells and macrophages) after intramuscular injection, for other applications of LNPs, uptake by specific cell types in specific organs in their natural location is required [65,66]. To solve this problem, Caitlin et al. proposed a targeted LNP (tLNP) strategy that can efficiently deliver mRNA to specific cell types based on the expression of cell surface markers [67]. These tLNPs are encapsulated with targeting antibodies, enabling mRNA cargo delivery to specific cells. In another example, mRNA could be delivered by ionizable lipid nanoparticles (iLNPs) to placental trophoblasts, endothelial cells, and immune cells in clinical practice [35]. Delivery of top-tier LNP formulations containing VEGF-A mRNA could induce placental vascular relaxation, indicating the potential of mRNA LNPs for protein replacement therapy in treating placental diseases during pregnancy [68]. LNPs deliver VEGFA mRNA into cells to enhance VEGFA expression and function. Treatment with LNP-VEGFA mRNA boosts the effectiveness of cardiac progenitor cells in promoting endothelial cell angiogenesis and modifying extracellular vesicles [69].

In general, researchers need to further enhance the drawbacks of the LNP delivery platform. The adoption of various novel approaches holds significant importance in augmenting mRNA delivery efficiency in vivo, thereby enhancing the survival rate and quality of life for cancer patients. 

#### 3.1.5. Self-Adjuvants and Adjuvants in Lipid-Based Nanoparticle Delivery Platforms

In the current research, in addition to lipid nanoparticle vaccines, adjuvant-combined lipid nanoparticle vaccines are being developed. Adjuvants typically function as immune stimulants that activate the innate immune response. The selection of appropriate adjuvants is crucial in the design of mRNA vaccines for different diseases. An optimal adjuvant for mRNA vaccines could stimulate a robust immune response and enhance the efficacy of the vaccine.

Recently, an mRNA vaccine was developed that contains lipids with cyclic amino groups that activate the STING (stimulator of interferon genes) pathway, which can trigger the stimulation of type I interferons (IFN) and potentially enhance adaptive immune responses [70,71]. Bowen et al. screened 480 biodegradable and ionizable lipids with cyclic amine adjuvants and used mRNAs encoding antigens with a natural adjuvant derived from the C3 complement protein to optimize the vaccine. This multi-adjuvant mRNA vaccine also increased antibody titers against SARS-CoV-2 about 10-fold. Compared with traditional mRNA transcript and LPS co-delivery strategies, the C3d adjuvant approach was more effective in inducing adaptive immune responses [26]. This multi-adjuvant mRNA vaccine system has the potential to enhance the safety, effectiveness, and convenience of mRNA vaccines for infectious diseases and other medical indications. LNPs with self-adjuvant properties enhance the innate immunity of mRNA-LNP vaccines. By partially replacing ionizable lipids with adjuvant-like lipids, not only is the delivery of mRNA enhanced, but the LNPs also exhibit acceptable tolerability and possess toll-like receptor 7/8-activating activity, thereby significantly enhancing the innate immunity of SARS-CoV-2 mRNA-LNP vaccines in mice [72]. In conjunction with the SARS-CoV-2 mRNA-LNP vaccine, this adjuvant-like lipid formulation has been optimized to thoroughly evaluate its innate and adaptive immune responses. The research indicated that it effectively activates DCs for antigen presentation, the expression of co-stimulatory molecules, and the production of specific cytokines (such as tumor necrosis factor-alpha), thus facilitating a transition from innate immunity to adaptive immunity [73].

Taken together, LNPs can overcome many challenges, as they have been demonstrated to enable efficient cellular uptake and potent mRNA delivery in vivo [74,75]. Currently, LNPs are the most clinically advanced non-viral drug delivery platform for nucleic acid therapeutics. Lipid-based vaccine delivery system design helps to improve the immunogenicity of the vaccine and antigen presentation efficiency, providing a potential new strategy for cancer immunotherapy.

### 3.2. Polymer-Based Nanoparticle Delivery Systems

The most recent research regarding polymeric nanoparticle materials includes research on polyamidoamine (PAMAM) dendrimers, polyethyleneimines (PEIs), poly(lactic-co-glycolic acid) (PLGA), poly(β-amino esters) (PBAEs), and cyclodextrins (CDs) [76,77]. Among these, the most widely investigated polymers for nucleic acid delivery are the PAMAM dendrimers and PEIs. PAMAM dendrimers are polymeric molecules with highly branched monomers emanating from a central core. Methyl acrylate and ethylenediamine are usually repeatedly added to the core according to the desired number of generations: G0, G1, G2, G3, G4, G5, etc. [78]. The negative charges on the DNA or RNA and the positive charges on the surface of the PAMAM interact with each other. This combination can form stable complexes for highly efficient transfection. PAMAMs can be functionalized easily by core modification and surface amendment. Different functional groups, such as NH_2_ groups, can be added to the terminus of the superficial branches of the PAMAM. In addition, other PAMAM end-group modifications, such as with amino acids, lipids, PEG, and other molecules, have been investigated to mitigate drawbacks such as toxicity and the lack of targeting [79]. The surface modification of PAMAM can been performed by neutralizing the charge through PEGylation. Many research studies have demonstrated that PEGylated PAMAM dendrimers enhance drug loading and drug release and decrease in vivo and in vitro cytotoxicity [80,81,82]. By conjugating a modified PAMAM dendrimer with a PEG–lipid, a replicon-RNA-based vaccine was developed that demonstrated protective immunity against multiple pathogens [83]. To improve targeting of the PAMAM delivery system, some ligands, including transferrin (Tf), angiopep-2, folic acid (FA), and low-density lipoprotein receptor-related protein 1 (LRP1), have been used to modify the PAMAM [84]. A FA-decorated PAMAM dendrimer exhibited greater tumor uptake and sustained highly localized retention in solid tumors [85]. 

Drugs interact with the polymer through electrostatic forces or covalent bonding, resulting in drug encapsulation within the polymer core or polymer matrix. This leads to enhanced drug stability, prolonged circulation time in vivo, reduced susceptibility to enzymatic and hydrolytic degradation, and improved drug bioavailability [86]. The cationic polymer can compress mRNA into nanoparticles with high density to obtain stable polyplexes that encapsulate the mRNA internally to enhance nucleic acid stability. Polymers are widely used for gene delivery as non-viral vectors. The current studies commonly utilize specific stimuli in vivo, such as GSH, pH, enzymes, ROS, ATP, etc., to monitor the cleavage of grafted degradable components within cationic polymers to maintain both blood stability and rapid intracellular degradation [87]. 

Although polymer-based mRNA delivery systems are not as advanced as lipid delivery systems in clinical applications, they have the potential to provide unique characteristics. Systems based on polymers have the capacity to create various nanostructures in an aqueous medium, have unique pharmacokinetic properties, and are easily lyophilized and stored for extended periods of time [88]. These characteristics could facilitate the development of sophisticated mRNA therapies. In a recent study, stimulating innate immune danger sensors, such as the cytosolic cyclic dinucleotide (CDN) sensor stimulator of interferon genes (STING), was shown to be a potent therapeutic strategy to generate durable anti-tumor immune responses. CDN-NPs have been established as a potent innate immune agonist treatment. Conjugation of CDN with NPs ensures higher effectiveness in terms of drug-loading capacity and stability in particle-mediated drug delivery. Ultimately, immune cells within secondary lymphoid organs and the tumor microenvironment (TME) absorb and transport CDN-NPs to proximal immune cells. The application of polymer nanoparticles increases the vaccine stability and payload, thereby expanding the therapeutic window in multi-gene tumor models and inhibiting tumor growth effectively [89]. This work underscores the importance of nanoparticle structure in modulating the efficacy of immunotherapy. Furthermore, a polymeric acid-based phosphatidyl polymer library has been developed for in vivo mRNA delivery with spleen-targeting ability in vivo, enabling effective mRNA delivery and potentially aiding the development of therapeutic or vaccine delivery systems targeting the spleen [90]. Alexandra et al. reported an inhalable polymer carrier for the delivery of therapeutic mRNAs to the lungs. They optimized biodegradable poly (amino ester) complexes using end-capping modifications and polyethylene glycol for mRNA delivery [91]. These complexes deliver mRNA to the lungs with high transfection efficiency, particularly in epithelial cells and antigen-presenting cells. This technology produced a mucosal vaccine against the severe acute respiratory syndrome SARS-CoV-2, and the findings indicated that intranasal administration of mRNA encoding spike protein complexes induced effective cellular and humoral adaptive immunity and protected susceptible mice from a lethal virus challenge. 

The surface of polymer nanocarriers can be easily modified with targeting ligands, making them an ideal choice for targeted mRNA delivery through ligand–receptor interactions, enhanced ligand-mediated mRNA delivery, and tumor targeting to enhance the precision of immunotherapy [89].

### 3.3. Protein-Based mRNA Delivery Systems

Protein nanoparticles are a type of nanoparticle system assembled from natural proteins or synthetic proteins by genetic engineering. Natural proteins such as serum proteins, hemoglobin, etc., are used in vaccine carrier design, while synthetic proteins can be custom-designed using genetic engineering techniques. Protein nanoparticles have a highly ordered and repetitive spatial structure and an optimal size for lymph node transport [92]. Such nanoparticles possess inherent antigenicity, allowing them to serve directly as vaccine antigens without the need for additional antigenic substances. Multiple protein-based delivery systems such as protamine, ferritin and antibody-drug conjugates have been extensively studied for RNA delivery. The arginine-rich protamine is a naturally occurring mixture of cationic peptides primarily utilized for the intracellular delivery of nucleic acids [93]. Due to its cationic property, protamine can spontaneously self-assemble with mRNA through electrostatic interaction and form stable complexes. Compared with naked mRNA, a protamine–mRNA complex can protect mRNA from ribonuclease (RNase)-mediated degradation [94]. But, protamine-based complexes show lower transfection efficiency. The addition of poly(acrylic acid) (PAA) derivates to protamine–RNA complexes could improve its transfectability with pH-dependent properties [93]. Nicole et al. demonstrated that electrostatic anti-CD33-antibody–protamine nanocarriers could transport their cargo safely into acute myeloid leukemia target cells [95]. Antibody–oligonucleotide conjugates (AOCs) are an antibody-mediated RNA delivery technology that has attracted increasing attention. The three parts of AOCs comprise an antibody-targeting cell surface receptor, a linker, and an oligonucleotide. Avidity Biosciences developed AOC1001, which is conjugated with a monoclonal antibody targeting the transferrin receptor 1 (TfR1) and siRNA [96,97]. In general, polypeptides have been conjugated to PEG to reduce protein adsorption and to increase their accumulation in tumors. Crowley reported a novel mRNA delivery system using polyacridine–PEG–polylysine that can reduce the degradation of mRNA by RNase metabolism [98].

Ferritin nanoparticles also have been used to serve as a new vaccine delivery platform that enhances vaccine stability and immunogenicity. The safety and immunogenicity of a novel ferritin-nanoparticle-based H2 influenza vaccine were assessed by Zhang et al. [99]. The study group evaluated the vaccine’s safety profile in human subjects, as well as its capacity to elicit robust immunological responses, including T cell activation and antibody accumulation. The utilization of this innovative ferritin nanoparticle vaccine technology could contribute to the development of a universal influenza vaccine and enhance pandemic preparedness. Furthermore, in clinical settings, the structure and composition of proteins can be adjusted during the preparation to achieve precise therapeutic effects.

Peptides with cationic or amphipathic groups (for instance, arginine) can be designed to effectively deliver mRNA molecules to cells. Peptide-based delivery systems show enhanced stability and protection by shielding the mRNA molecules from degradation by nucleases and other cellular and extracellular enzymes, thereby effectively improving the efficacy of mRNA vaccines. Importantly, peptide-based delivery systems utilizing naturally synthesized peptides can reduce adverse reactions and enhance safety. Repetitive arginine–alanine–leucine–alanine (RALA), a cell-penetrating peptide, possesses the capability of delivering mRNA to dendritic cells and inducing T cell-mediated immunity [100]. CoVac-1, an epitope-based peptide T cell activator targeting SARS-CoV-2, has demonstrated favorable safety and efficacy in clinical trials by eliciting robust and efficient T cell responses [101]. Studies indicate that CoVac-1 exhibits a promising safety profile in B cell/antibody-deficient patients, enabling their participation in clinical Phase III assessments of their safety and effectiveness. Xu et al. has also developed an optimized polyethylene-glycolated peptide with enhanced mRNA delivery efficiency and improved safety in mice [102]. Moreover, cationic peptides as candidate carriers for mRNA have been shown to efficiently deliver nucleic acids to eukaryotic cells. A cationic peptide-based mRNA nanoparticle could induce efficient antigen-specific CD8^+^ T cell responses [103]. 

Above all, protein-based mRNA delivery systems show promise in the field of gene therapy, enabling the delivery of therapeutic mRNA to correct genetic disorders. They are also being used for mRNA-based vaccines to induce immune responses against infectious diseases or cancer. Although polypeptide vaccines possess the advantages of high safety and strong specificity, they currently face challenges such as weak immunogenicity, necessitating multiple immunizations, and inconsistent immune responses. In the future, researchers should strive to enhance and advance polypeptide vaccines through epitope design integration, carrier optimization, and the use of novel adjuvants. These advancements are pivotal in the realm of modern vaccine development.

### 3.4. Other Formulations Used in mRNA Delivery

#### 3.4.1. Virus-like Particles (VLPs) 

Virus-like particles (VLPs) are composed of recombinant proteins expressing viral surface antigens, mimicking the structure of real viruses but lacking viral nucleic acids [104]. VLPs can accurately replicate the structure of viruses, triggering robust immune responses. The delivery of mRNA to specific cell types is facilitated by a novel approach utilizing VLPs, which encompasses the essential structural proteins necessary for viral capsid assembly. VLPs can be engineered to encapsulate and transport targeted mRNA molecules. Recent research indicates that VLPs are generated through the fusion of nucleocapsid proteins with the MS2 coat protein (MCP), enabling recruitment of MS2 hairpin-containing mRNAs into the VLPs [105]. Cai et al. developed a novel VLP-mRNA delivery system by utilizing the principle of the mRNA stem–loop structure and specific recognition of the phage capsid protein, which effectively mitigates or even eliminates off-target effects [106]. VLPs are very safe, as they do not cause infection or replication, making them suitable for use as vaccine carriers. 

Furthermore, VLPs can display multivalent antigenic epitopes on their surface, which can extensively cross-link B cell receptors (BCRs), activate B cells, and induce a robust and long-lasting antibody response due to their highly repetitive and rigid structure [107]. According to Chang et al., VLPs have the ability to elicit potent antibody responses [108]. Viral-like nanoparticle delivery systems have demonstrated their significance in TLR7 signaling, particularly in B cells. They could facilitate early germinal center formation and affinity maturation while being responsible for generating and preserving BCR diversity. However, the expression of conventional mRNA vaccines may be hindered by the presence of viral surface antigens on the cell membrane. Researchers have presented encoding self-assembling enveloped virus-like particles (eVLPs) with the features of mRNA and protein nanoparticle-based vaccines [109]. The assembly of eVLP is accomplished by combining the ESCRT with the ALIX binding region (EABR) (the inner body sorting complex required for transport) of the cytoplasmic tail of SARS-CoV-2 spike protein. eVLP-ESCRT mRNA vaccination can have the effect of both eVLPs and mRNA vaccine, yielding effective CD8^+^ T cell responses and superior neutralizing antibody responses with only one vaccination. This system can enhance the potency and breadth of vaccine-induced responses and enable longer-lasting protection against SARS-CoV-2 and other viruses.

In summary, VLP-based vaccines have achieved success and are being used in the market. In the future clinical practices, VLPs can be designed to carry different types of antigens including against the current influenza virus HPV, hepatitis B, and so on to against the diseases [109].

#### 3.4.2. Inorganic-Based Nanoparticles 

Inorganic nanoparticles play a crucial role in nucleic acid delivery and imaging. Similar to gold nanoparticles, silica nanoparticles and iron oxide nanoparticles are inorganic nanoparticles. Inorganic nanoparticles have a large surface area, enabling them to carry a large quantity of mRNA molecules, thereby enhancing delivery efficiency and stability [110]. They have the characteristics of a dense structure which can protect mRNA molecules from degradation and prolong their presence in the body to improve delivery efficiency. At the same time, they can form rational particle sizes to facilitate the direct transport of antigens to lymph nodes, promoting antigen presentation and enhancing adaptive immune responses. For example, gold nanoparticles were used for DNA and siRNA delivery. Furthermore, its surface can be modified by anionic nucleic acids, polycations, and targeting ligands [111]. To prevent pneumonic plague, Yang et al. developed a novel mesoporous silica manganese nanoparticle (AMMSN) loaded with rF1-V10 (rF1-V10@AMMSN) [112]. Subcutaneous vaccination with rF1-V10@AMMSN using a prime–boost technique induced the robust production of IgG antibodies specific to rF1-V10. Moreover, in vivo uptake of rF1-V10@AMMSN by DCs triggered the cyclic GMP-AMP synthase (cGAS)-stimulator of interferon genes (STING) pathway and facilitated DC maturation, leading to the generation of IFN-I. The limitations of inorganic-based nanoparticles as delivery vehicles include the insufficient co-delivery of antigens and adjuvants and inadequate immune stimulation. To modulate the interaction between inorganic nanoparticles and cells, surface modification could facilitate targeted delivery and intracellular release. By adjusting the structure and properties of inorganic nanoparticles, controlled release of mRNA can be achieved, enhancing delivery precision and efficiency.

#### 3.4.3. Cationic Nanoemulsions

Cationic nanoemulsions (CNEs) have been proposed as a potential delivery system for nucleic acids [113,114]. They are composed of oil droplets dispersed in an aqueous phase, stabilized by positively charged surfactants. The positive charge of CNEs facilitates interactions with negatively charged cell membranes, promoting cellular uptake of the mRNA cargo and enhancing delivery efficiency. CNEs can efficiently encapsulate mRNA molecules within their oil droplets, protecting the mRNA from degradation and improving its stability during delivery [114]. Surface modifications of CNEs can enable the targeted delivery of mRNA vaccines to specific cells or tissues, enhancing therapeutic efficacy and reducing off-target effects. Luis et al. developed a CNE delivery system to deliver a self-amplifying mRNA vaccine based on the MF59 adjuvant, which elicited a potent immune response [115]. Some CNEs exhibit intrinsic adjuvant properties; for instance, activating the immune system and enhancing the immune response to mRNA vaccine antigens. Additionally, Montanide ISA 51 and Montanide ISA 720 are being evaluated in clinical trials as adjuvants for influenza, malaria, melanoma vaccine, and other diseases [116]. The CNE products offer several advantages compared with the conventional systems currently utilized, which can effectively meet the existing requirements in gene therapy clinical practice.

## 4. Nanovaccines and Anti-Tumor Immunity

Nanoparticle delivery platforms serve as a promising tool in vaccine development, especially in enhancing anti-tumor immunity and improving immune responses against cancer cells. Tumor-specific antigens can be efficiently delivered to antigen-presenting cells (such as DCs) through nanoparticle delivery systems, thereby enhancing antigen presentation and stimulating CD8^+^ T cells. Sustained and high-affinity antibodies, T follicular helper cell (Tfh) responses, and GC formation could be elicited by nucleoside-modified mRNA-lipid nanoparticle (mRNA-iLNP) vaccines [117]. The lipid nanoparticle (SMART-LNP) system targeting the mannose receptor was created by Gokulnath et al. to improve antigen presentation to DCs [118]. This method was able to transport mRNAs into DCs in an efficient manner. The nanovaccines were also capable of eliciting a robust T cell immune response. By selectively targeting CD206 on DCs, STLNPs-Man was able to effectively downregulate the expression of CTLA-4 on the surface of CD8^+^ T cells, thereby augmenting the cytotoxic potential of activated CD8^+^ T cells. Relatively low dosages of mRNA were sufficient to achieve complete tumor eradication when combined with the immune checkpoint inhibitor αPD-L1 [64,119]. The OspA mRNA-LNP vaccine might work as a successful Lyme disease preventive strategy. Research indicated that a robust infiltration of innate immune cells and the induction of antigen-specific CD4^+^ and CD8^+^ T cell responses were observed in mice following administration of the OspA mRNA-LNP vaccine [120].

The continuous release of antigens from nanoparticles can enhance immune cell activation and augment anti-tumor immune responses by prolonging the exposure of immune cells to antigens. In vaccination strategies, maternal antibodies (matAbs) contribute to the suppression of infant immune responses in clinical settings. Scientists speculate that the nucleoside-modified mRNA-LNP vaccine’s ability to partially overcome matAb suppression and to induce a prolonged germinal center response may be attributed to its sustained antigen expression. According to the studies, enhanced germinal center responses are associated with the prolonged availability of antigens. Recent research suggests that B cells primed with antigens exhibit more efficient infiltration into germinal centers compared with naïve B cells [121]. All things considered, activated monocytes, macrophages, and DCs are the primary cells responsible for the protein synthesis, antigen presentation, and efficient mRNA-iLNP absorption in lymphoid tissues that enhance adaptive immune responses. Nanoparticles can selectively target specific immune cells, thereby enhancing the specificity and efficacy of vaccinations through the formation of tertiary lymphoid structures (TLSs). These findings suggest that the materials used in the preparation of the nanovaccine are biocompatible, cost-effective, and FDA-approved, and that the nanovaccine can promote TLS formation, significantly enhance local immune responses and delay tumor growth, indicating good clinical prospects for nanovaccine applications. Furthermore, incorporating CpG and Mn2^+^ into the nanovaccine strongly activates T cells and B cells, stimulates DC expression of LT-α and CCL21, and induces peripheral lymph node addressin protein (PNAd) vascular formation [122,123].

Nanoparticles can also be designed to modulate immune responses based on clinical needs, such as promoting the activation of cytotoxic T cells, enhancing the production of pro-inflammatory cytokines, and overcoming immune tolerance mechanisms in the tumor microenvironment. As a contrast agent, iron oxide nanoparticles (IONPs) were applied in magnetic resonance imaging in a clinical application [124]. Chen et al. found that IONPs significantly increased the production of STING agonist MSA-2, thereby promoting the generation of IFN-I [125]. The study revealed the potential immunostimulatory role of IONPs in STING cascade activation, providing a scalable and easily translatable strategy for personalized cancer vaccine immunotherapy. Another research study demonstrated that the adjuvant autogene cevumeran, a personalized novel antigen vaccine based on uridine mRNA–lipid nanoparticles, was safe and feasible when administered in conjunction with atezolizumab and mFOLFIRINOX. Furthermore, it exhibited the ability to induce a significant population of neoantigen-specific T cells in 50% of patients diagnosed with unresectable pancreatic ductal adenocarcinoma (PDAC) [126].

Overall, nanovaccines hold great potential for leveraging nanotechnology to improve anti-tumor immunity and to enhance the effectiveness of cancer immunotherapy [33,127]. Current research in this field aims to optimize the design and delivery of nanovaccines to achieve personalized and effective cancer treatment strategies.

## 5. Conclusion and Future Perspectives

Nanoparticle-based delivery platforms have been demonstrated to be revolutionary as vaccine vehicles with high delivery efficacy. The status of the current research on nanoparticle-based vaccine platform development shows that there is growing interest in exploring the potential of nanoparticles as vaccine delivery systems. Researchers are actively investigating various types of nanoparticles, including lipid, polymer, protein and inorganic nanoparticles, for the delivery of antigens and adjuvants to the immune system. One area of active research is the development of novel nanoparticle formulations that can enhance the stability and immunogenicity of vaccines. Now, lipid nanoparticles and other nanoparticles nanoparticles for nucleic acids, especially mRNA delivery systems, have shown broad potential for vaccine development. Lipid nanoparticles have also shown potential for the efficient delivery of antigens to immune cells in preclinical and clinical studies. In this review, we focused on the application prospect of mRNA vaccines, especially lipid-nanoparticle-based delivery platforms, as well as the design of polymeric nanoparticles that can efficiently deliver antigens to immune cells. Additionally, the use of inorganic nanoparticles such as gold and iron oxide nanoparticles for vaccine delivery is also a focus of ongoing research. 

Furthermore, researchers are also exploring the potential of nanoparticles to enable targeted vaccine delivery to specific immune cells or tissues. This involves the surface modification of nanoparticles with targeting ligands that can selectively bind to receptors on immune cells, thereby enhancing uptake of the vaccines and the activation of immune responses. In addition to vaccine delivery, current research efforts are also focused on understanding the immunological mechanisms underlying the interaction between nanoparticles and the immune system. The nanoparticle delivery platform effectively delivers tumor-specific antigens to antigen-presenting cells (such as DCs), thereby enhancing antigen expression and activating CD8+ T cells. Also, nanoparticles can be designed to target specific immune cells or tumor microenvironments, such as to form TLSs, increasing the specificity and effectiveness of vaccines. Nanoparticle delivery platforms serve as a promising tool in vaccine development, especially in enhancing anti-tumor immunity and improving immune responses against cancer cells. Although T cells have the ability to destroy tumors, cancer cells can gain ways of evading immune attacks. Among these barriers, immunosuppressive cells and intrinsic resistance account for the failure of many therapeutic cancer vaccines [128]. There are some ways to overcome these barriers for cancer vaccines to achieve their potential. For instance, combining cancer vaccines with immune checkpoint inhibitors that block PD-1 binding to PD-L1 and other treatment regimens could improve the anti-tumor efficacy of vaccines.

Nanoparticle vaccines have a significant effect on genetic diseases, infectious diseases, and cancer treatment [129]. In particular, various types of nanoparticles have been used for mRNA vaccines for cancer treatment. In recent years, mRNA vaccine technologies have received increasing research and commercial attention because of their potential in the prevention of infectious diseases and cancer treatment. Over the past few years, nanoparticle-based vaccines have emerged as a primary tool in the field of prophylactic vaccines; now, they are being tested in oncology clinical trials. Numerous therapeutic cancer vaccines have shown promising results in clinical trials, including for melanoma, pancreatic cancer, and lymphoma. A personalized mRNA vaccine for pancreatic cancer can trigger neoantigen-specific T cells in about 50% participants developing cancer [126]. Although mRNA vaccines are showing encouraging results in trials, there are many obstacles ahead. The future development and clinical trials of therapeutic cancer vaccines may be shaped by several factors, including unwieldy clinical trials, the need for immunity monitoring such as T-cell monitoring, and the need to streamline production. 

Nanoparticle delivery platforms elevate the level of engineering and control by fine-tuning parameters such as drug dissolution, diffusivity, half-life, toxicity, pharmacokinetics, and biodistribution. In recent years, numerous studies have demonstrated the multifunctional capabilities of nanoparticles as sensors, drug carriers, and diagnostic agents. The rationale behind the utilization of nanoparticles as drug delivery systems is based on at least three mechanisms: (1) augmenting the infiltration and retention of nanoparticles within solid tumors; (2) facilitating the transportation of insoluble drugs in the bloodstream through a stable colloidal system; (3) enabling the controlled release of encapsulated drugs. However, as a delivery platform, the potential toxicity of nanoparticles themselves and the stability of the encapsulated mRNA pose urgent challenges for numerous researchers. Triggering a tumor-specific cytotoxic T cell response remains an important challenge in cancer immunotherapy. Optimally selecting a delivery system is crucial for vaccine applications in cancer treatment or infectious diseases. Simultaneously, enhancing the specificity of nanoparticle targeting toward cells or organs remains crucial in order to provide broader technical support for the next generation of cancer vaccines.

In summary, nanoparticle-based vaccine platforms have shown potential as delivery vehicles in the prevention of cancer treatment and infectious diseases. In the future, we hope to see an increased application of vaccines using various targeting approaches or combination therapy, which will allow this versatile and potent platform to gain further new therapeutic applications in oncology.

## Figures and Tables

**Figure 1 vaccines-12-00727-f001:**
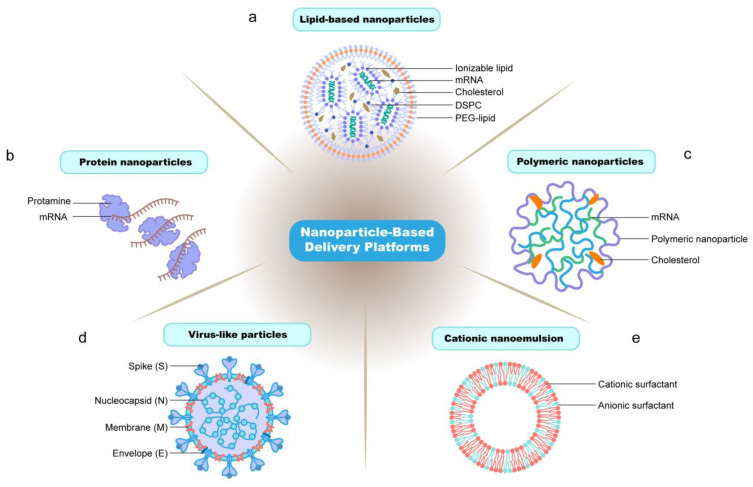
The main nanoparticle-based delivery platforms for mRNA vaccines. Description of the general structure of the nanoparticle platforms and their basic components. Commonly used delivery methods and carrier molecules for mRNA vaccines are shown: (**a**) lipid-based nanoparticles; (**b**) protein nanoparticles; (**c**) polymeric nanoparticles; (**d**) virus-like particles; (**e**) cationic nanoemulsion. (**a**) Lipid-based nanoparticles encapsulate mRNA in their core. They consist of three components: ionizable lipids, cholesterol, and helper lipids such as DSPC and PEG-lipids. (**b**) Protein nanoparticle systems utilizing naturally synthesized peptides such as protamine. (**c**) Polymers, such as polymeric nanoparticles and cholesterol, form polymer–mRNA complexes. (**d**) A typical virus-like vaccine based on four structural proteins: spike (S), envelope (E), membrane (M), and nucleocapsid. (**e**) The cationic nanoemulsion shell is made up of cationic lipid and anionic surfactants, and the mRNA adsorbs to the surface via electrostatic binding.

**Figure 2 vaccines-12-00727-f002:**
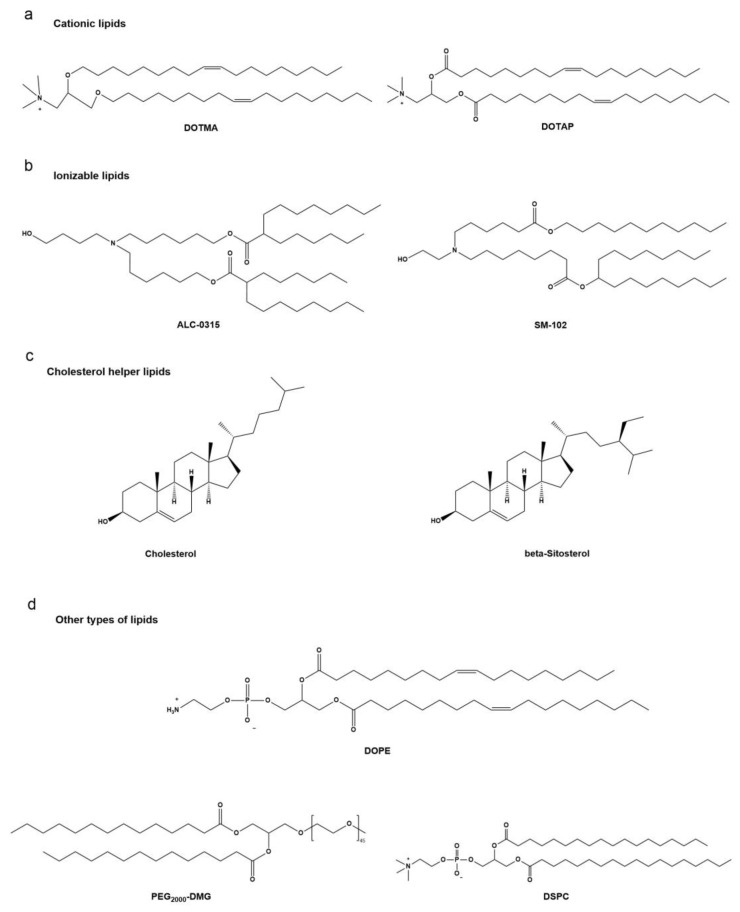
Chemical structures of lipids and lipid derivatives for nucleic acid delivery.

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
