# Peer review of "An Overview of Nanoparticle-Based Delivery Platforms for mRNA Vaccines for Treating Cancer"

_vaccines, 2024, doi:10.3390/vaccines12070727_

Round 1

Reviewer 1 Report

Comments and Suggestions for Authors

The paper begins by outlining the significance of nanoparticle-based delivery platforms for vaccines, emphasizing lipid nanoparticles (LNPs) and their application in mRNA vaccine delivery. The introduction provides a comprehensive overview of the evolution and importance of vaccination and the specific advantages of using nanoparticles, such as enhanced immunogenicity and stability.

Strengths:

1. The paper thoroughly examines various types of nanoparticles, including lipids, polymers, proteins, and inorganic nanoparticles, offering a broad perspective on the field.

2. It highlights the clinical applications of these technologies, particularly in the context of the COVID-19 pandemic, demonstrating the research's timely and practical significance.

3. The authors provide deep mechanistic insights into how nanoparticles enhance vaccine delivery and efficacy, which could be invaluable for readers and researchers looking to understand or utilize these delivery systems.

Drawbacks:

1. While the paper discusses the efficacy and immediate immunological responses to nanoparticle-based vaccines, there is limited discussion on their long-term safety and potential side effects, which are crucial for clinical applications.

2. The paper could benefit from more direct comparisons between the different types of nanoparticles, discussing their benefits and their relative limitations in a comparative context.

3. Although the paper covers a range of nanoparticle systems, it does not significantly highlight novel or groundbreaking discoveries in the nanoparticle vaccine field. Adding unique case studies or newly developed technologies could enhance its impact.

Recommendations:

1. Incorporating comparative studies or meta-analyses could provide clearer guidance on choosing between nanoparticle systems based on specific vaccine requirements, patient populations, or disease targets.

2. Highlighting novel research or recent breakthroughs in nanoparticle technology could make the paper more appealing to a broader audience, including those involved in cutting-edge research.

Author Response

Response to Reviewer 1 Comments

1. Summary

Thank you very much for taking the time to review this manuscript. We are so grateful for your recommendation. Please find the detailed responses below and the corresponding revisions/corrections highlighted in the re-submitted files.

2. Point-by-point response to Comments and suggestions for Authors.

Comments 1: Incorporating comparative studies or meta-analyses could provide clearer guidance on choosing between nanoparticle systems based on specific vaccine requirements, patient populations, or disease targets.

Response 1: Thank you for your recommendation. We agree with this comment. We have tried our best to revise the manuscript according to your kind and constructive comments and suggestions. We appreciated for reviewers’ warm work earnestly, and hope that the connection will meet with approval. Once again, thank you very much for your comments and suggestions.

Comments 2: Highlighting novel research or recent breakthroughs in nanoparticle technology could make the paper more appealing to a broader audience, including those involved in cutting-edge research.

Response 2: Thank you very much for your comments and suggestions. We have tried our best to revise the manuscript according to your kind and constructive comments and suggestions. We sincerely hope that this revised manuscript has addressed all your comments and suggestions.

Reviewer 2 Report

Comments and Suggestions for Authors

The authors of the review highlighted the achievements in the development of Nanoparticle-Based Delivery Platforms for Vaccines over the past few years.

The review article is not written to the point; there are many repetitions that increase the volume. It is necessary to remove repetitions and add one figure to each type of nanoparticle with chemical formulas

please devote one paragraph to delivery systems for CAR-T therapy.

1. (line 91) the title «2.1. Normal tumor vaccine types» must be changed. There are abnormal types? Traditional?

2. (line 97) Please add the abbreviation (DC) for dendritic cells when first appearing (not in line 119).

3. (line 100) Please add how TPOP stands for

4. (line 102) Please add doxorubicin for the abbreviation DOX

5. (line 108) the target (new antigens arise from mutations in tumor cells….) is described for tumor neoantigen vaccines, but not the active substance (peptide, protein, lysate?). Please add

6. (line 140) In host cells, mRNA encoding antigen are synthesized and then triggerS immune responses

7. (line 170) There is no subject and predicate in the sentence «However, due to the inherent structural characteristics of mRNA, such as large molecular weight, high negative charge, and susceptibility to degradation». It's better to combine it with the next sentence.

8. Figure 1

a) «Commonly used delivery methods and carrier molecules for mRNA vaccines are shown: peptide-based delivery lipid-based nanoparticles (a) protein nanoparticles (b) polymeric nanoparticles (c) virus-like particles (d) cationic nanoemulsion (e).»

Numbering does not match

b) (line 183) A typical virus-like VACINNES based on four structural proteins:

9. (line 187) Therefore, efficient delivery of antigen-encoding mRNA that has been transcribed in vitro into cells is a key challenge in RNA cancer vaccine therapy

10. (line 198) Here, we will DESCRIBE several common nanoparticle 198 delivery platforms with their current research status.

11. (line 214) «Additionally, LNPs demonstrate excellent stability which effectively protect antigens within the carrier from degradation and damage».

It seems to me that this paragraph is about the antigen-encoding mRNA, and not about the antigen itself; especially considering the reference [40] above

Please correct

12. (line 229) negatively charged

13. (line 269) Disadvantages and optimization of lipid-based nanoparticle delivery platform

14. (line 356) «Although polymer-based mRNA delivery systems are not as advanced in clinical applications as lipid delivery systems they have the potential to provide unique characteristics. And there also have challenges such as relatively lower transfection efficiency and potential toxicity. For example, polymer-based systems can assemble various nanostructures under aqueous conditions with the ability for lyophilization and long-term storage, and exhibit unique pharmacokinetics [76].»

Please explain how these sentences are logically connected or paraphrase

15. (line 363) what CDN stands for?

16. (line 375) And Alexandra et al. re- 375 ported an inhalable polymer carrier for delivering therapeutic mRNA to the lungs.

Delete AND at the begining

17. (line 395) Arginine-rich protamine is natural cationic peptide mixture mostly known used for cellular delivery of nucleic acid

Please paraphrase

The sentence is unclear

18. (line 408) (for instance, argiNine)

19. (line 419) cLINical Phase III safety

20. (line 430) Virus-like particles (VLPs) are composed of recombinant proteins expressing viral surface proteins, mimicking the structure of real viruses but lacking viral nucleic acids

«proteins expressing viral surface proteins»?

21. (line 436) what BCR stands for?

22. (line 478) have been proposed

23. Additionally, Montanide ISA 51 and Montanide ISA are being evaluated in clinical trials as adjuvants for influenza, malaria, melanoma, and other cancer vaccines

Delete other (influenza, malaria, melanoma vaccines are not cancer vaccines)

Comments on the Quality of English Language

Extensive editing of English language required

some sentences are difficult to understand and some are not related in meaning

Author Response

Response to Reviewer 2 Comments

1. Summary

We sincerely thank you for thoroughly examining our manuscript and providing very helpful comments to guide our revision. We have tried our best to revise the manuscript according to your kind and constructive comments and suggestions. To be more clearly and in accordance with the reviewer concerns, we have added some details about delivery systems for CAR-T therapy on page section 3.1.2 and the chemical formulas in Figure 2. We also have provided a point-by-point responses to the comments. We sincerely hope that this revised manuscript has addressed all your comments and suggestions. We appreciated for reviewers’ warm work earnestly, and hope that the connection will meet with approval. Once again, thank you very much for your comments and suggestions.

2. Point-by-point response to Comments and Suggestions for Authors

Comments 1: (line 91) the title «2.1. Normal tumor vaccine types» must be changed. There are abnormal types? Traditional?

Response 1: Thank you for pointing this out. We have made correction according to the Reviewer’s comments. We change the title to “Traditional tumor vaccine types” in line 88.

Comments 2: (line 97) Please add the abbreviation (DC) for dendritic cells when first appearing (not in line 119).

Response 2: Thank you for pointing this out. The incorrect writing on our part is deeply regretted, I have changed the abbreviation (DC) for dendritic cells in line 74.

Comments 3: (line 100) Please add how TPOP stands for.

Response 3: Thank you for your recommendation. The TPOP is an in situ nanovaccine which is based on nanoparticles that regulate lipid metabolism and stimulate innate immune response was developed by Qin et al. And we have explained in the section 2.1 text in the line 97 to 100.

Comments 4: (line 102) Please add doxorubicin for the abbreviation DOX.

Response 4: Thank you for your reminder. We were really sorry for our careless mistakes. Thank you for your reminder in line 101.

Comments 5: (line 108) the target (new antigens arise from mutations in tumor cells….) is described for tumor neoantigen vaccines, but not the active substance (peptide, protein, lysate?). Please add.

Response 5: Thank you for your comments, we have modified the contents according to your requirements, and added the suggested content to the manuscript on line 105 to 111. I hope you can be satisfied.

Comments 6: (line 140) In host cells, mRNA encoding antigen are synthesized and then trigger S immune responses.

Response 6: Thank you for your recommendation. Thank you for pointing this out. The syntax error has been corrected in section 2.2.

Comments 7: There is no subject and predicate in the sentence «However, due to the inherent structural characteristics of mRNA, such as large molecular weight, high negative charge, and susceptibility to degradation». It's better to combine it with the next sentence.

Response 7: Thanks for your suggestion. We have tried our best to polish the language in the revised manuscript. We have been changed statement expression in the line 156-159.

Comments 8: Figure 1

a) «Commonly used delivery methods and carrier molecules for mRNA vaccines are shown: peptide-based delivery lipid-based nanoparticles (a) protein nanoparticles (b) polymeric nanoparticles (c) virus-like particles (d) cationic nanoemulsion (e).»

Numbering does not match.

b) (line 183) A typical virus-like VACINNES based on four structural proteins.

Response 8: We were really sorry for our careless mistakes. Thank you for your reminder. The number has been moved forward and the error has been corrected in the legends of Figure 1.

Comments 9: (line 187) Therefore, efficient delivery of antigen-encoding mRNA that has been transcribed in vitro into cells is a key challenge in RNA cancer vaccine therapy.

Response 9: Thank you for your help very much. We have corrected the error in line 172-173.

Comments 10: (line 198) Here, we will DESCRIBE several common nanoparticle 198 delivery platforms with their current research status.

Response 10: Thank you for your recommendation. We have changed “introduce” to “describe” in the main text in line 182.

Comments 11: (line 214) «Additionally, LNPs demonstrate excellent stability which effectively protect antigens within the carrier from degradation and damage».

It seems to me that this paragraph is about the antigen-encoding mRNA, and not about the antigen itself; especially considering the reference [40] above.

Please correct.

Response 11: Thank you for pointing out this. We missed that one detail and have revised in the line 221-223.

Comments 12: (line 229) negatively charged.

Response 12: Thank you for your reminder. We have revised this in line 236.

Comments 13: (line 269) Disadvantages and optimization of lipid-based nanoparticle delivery platform.

Response 13: Thank you for pointing this out. We have changed the title of 3.1.3.

Comments 14: (line 356) «Although polymer-based mRNA delivery systems are not as advanced in clinical applications as lipid delivery systems they have the potential to provide unique characteristics. And there also have challenges such as relatively lower transfection efficiency and potential toxicity. For example, polymer-based systems can assemble various nanostructures under aqueous conditions with the ability for lyophilization and long-term storage, and exhibit unique pharmacokinetics [76]».

Please explain how these sentences are logically connected or paraphrase.

Response 14: Thanks for your suggestion. We feel sorry for our poor writings. We tried our best to improve the manuscript and made some changes to these sentences in the line 429-431.

Comments 15: (line 363) what CDN stands for?

Response 15: The CDN is stand for the cytosolic cyclic dinucleotide. And we have explained in the revised text in the line 432-434.

Comments 16: (line 375) And Alexandra et al. re- 375 ported an inhalable polymer carrier for delivering therapeutic mRNA to the lungs. Delete AND at the beginning.

Response 16: Thank you for your suggestion. We have deleted the word in line 447.

Comments 17: (line 395) Arginine-rich protamine is natural cationic peptide mixture mostly known used for cellular delivery of nucleic acid.

Please paraphrase.

The sentence is unclear.

Response 17: We feel sorry for our unclear writings. We tried our best to improve the manuscript and made some changes to these sentences in the line 467-469.

Comments 18: (line 408) (for instance, argiNine).

Response 18: We were really sorry for our careless mistakes. Thank you for pointing it out. We have corrected it in line 483.

Comments 19: (line 419) cLINical Phase III safety.

Response 19: We feel so sorry for our carelessness. We have corrected it in our revised manuscript in line 494.

Comments 20: (line 430) Virus-like particles (VLPs) are composed of recombinant proteins expressing viral surface proteins, mimicking the structure of real viruses but lacking viral nucleic acids

«proteins expressing viral surface proteins»?

Response 20: Thank you for your recommendation. We feel sorry for our carelessness. We have corrected our expression in the line 512-514 and we also feel great thanks for your point out.

Comments 21: (line 436) what BCR stands for?

Response 21: BCR is stand for B cell-receptor. And we have explained in the revised manuscript in line 525.

Comments 22: (line 478) have been proposed.

Response 22: We have carefully checked the manuscript and corrected the errors accordingly in line 570.

Comments 23: Additionally, Montanide ISA 51 and Montanide ISA are being evaluated in clinical trials as adjuvants for influenza, malaria, melanoma, and other cancer vaccines.

Delete other (influenza, malaria, melanoma vaccines are not cancer vaccines).

Response 23: Thank you for pointing this out. We have changed the expression in the line 582-583.

4. Response to Comments on the Quality of English Language

Comment 1: Extensive editing of English language required.

Response 1: Thank you for thoroughly examining our manuscript. We have re-edited the English language of our manuscript.

Comment 2: Some sentences are difficult to understand and some are not related in meaning

Response 2: Thank you for your helpful comments. We have tried our best to revise the manuscript to be more clearly according to your kind and constructive comments and suggestions.

Reviewer 3 Report

Comments and Suggestions for Authors

A small review with many redaction and grammar errors. All the sections in the review must be broaden, not only to include more details (as described below), but to include more references. The author are just summarizing a few works. 

Change title to: "An Overview of Nanoparticle-Based Delivery Platforms for mRNA Vaccines to Treat Cancer"

Line 207-208, "and other types of lipids such as phospholipids, cholesterol or polyethylene glycol (PEG)", PEG is not a lipid

Line 212, "The surface of LNPs is easily modifiable", properly describe how this is accomplished

Line 212-213, "Surface modifications could usually improve the targeting property of nanoparticles", properly describe how to target the LNP-based vaccines

Section 3.1.1, how to prepare LNP must be included in this section.

Line 215, "LNPs demonstrate excellent stability", if this were the case (at least for the COVID-19 vaccines) the storage temperature required of -80 C (at least at the beginning of the pandemic) would have been unnecessary. LNPs are unstable. Include in this section how are the LNPs stabilized, electrostatically or sterically?

Section 3.2, please describe how the delivery systems are prepared. How they are modified for targeting. How they are stabilized. The role of PEG?

Line 408, "argine"?

Section 3.3, please describe how the delivery systems are prepared. How they are modified for targeting. How they are stabilized. The role of PEG?

Section 3.4.1, how is the mRNA loaded into VLPs?

Line 457, "Not only they can encapsulate mRNA molecules", gold nanoparticles cannot encapsulate anything

Section 5. The main drawbacks of using nanoparticles are their stability and the toxicity, both aspects need to be addressed in this section.  

Comments on the Quality of English Language

A native English speaker must review this manuscript. 

Author Response

Response to Reviewer 3 Comments

1. Summary

Thank you very much for taking the time to review this manuscript. Please find the detailed responses below and the corresponding revisions/corrections highlighted in the re-submitted files.

2. Point-by-point response to Comments and Suggestions for Authors

Comments 1: Change title to: “An overview of Nanoparticle-Based Delivery Platforms for mRNA Vaccines to Treat Cancer”.

Response 1: Thank you for your recommendation. We have changed the title in accordance with your suggestion.

Comments 2: Line 207-208, "and other types of lipids such as phospholipids, cholesterol or polyethylene glycol (PEG)", PEG is not a lipid.

Response 2: Thanks for your careful checks. We are sorry for our carelessness. Based-on your comments, we have made the corrections within the whole manuscript in line 193.

Comments 3: Line 212, "The surface of LNPs is easily modifiable", properly describe how this is accomplished.

Response 3: Thanks for your great suggestion on improving the accessibility of our manuscript. We have added “how the surface of LNPs modifiable” in the line 213-219.

Comments 4: Line 212-213, "Surface modifications could usually improve the targeting property of nanoparticles", properly describe how to target the LNP-based vaccines.

Response 4: The reviewer's suggestion has prompted us to incorporate additional explanations to substantiate this concept in the line 209-213.

Comments 5: Section 3.1.1, how to prepare LNP must be included in this section.

Response 5: We sincerely appreciate the valuable comments. We have added “how to prepare LNP” in the line 193-202.

Comments 6: Line 215, "LNPs demonstrate excellent stability", if this were the case (at least for the COVID-19 vaccines) the storage temperature required of -80 C (at least at the beginning of the pandemic) would have been unnecessary. LNPs are unstable. Include in this section how are the LNPs stabilized, electrostatically or sterically?

Response 6: We thank the reviewer for pointing this out. We missed that one detail and have revised in line 222-223.

Comments 7: Section 3.2, please describe how the delivery systems are prepared. How they are modified for targeting. How they are stabilized. The role of PEG?

Response 7: We eagerly anticipate receiving your insightful comments. The inclusion of preparation methods and the elucidation of PEG's role have been incorporated into this section IN LINE 419-421.

Comments 8: Line 408, "argine".

Response 8: Thanks for your careful checks. We are sorry for our carelessness. We have corrected it in line 483.

Comments 9: Section 3.3, please describe how the delivery systems are prepared. How they are modified for targeting. How they are stabilized. The role of PEG?

Response 9: We sincerely appreciate the valuable comments. We have added some details in the line 469-472.

Comments 10: Section 3.4.1, how is the mRNA loaded into VLPs?

Response 10: Thanks for your great suggestion on improving the accessibility of our manuscript. We have added “how is the mRNA loaded into VLPs” in the line 515-523.

Comments 11: Line 457, "Not only they can encapsulate mRNA molecules", gold nanoparticles cannot encapsulate anything".

Response 11: Thank you for pointing this out. We have changed the expression approach in line 550-551.

Comments 12: Section 5. The main drawbacks of using nanoparticles are their stability and the toxicity, both aspects need to be addressed in this section.

Response 12: We think this is an excellent suggestion. We have re-written this part according to the Reviewer’s suggestion. In Section 3.5

4. Response to Comments on the Quality of English Language

Point 1: A native English speaker must review this manuscript.

Response 1: Thank you for thoroughly examining our manuscript. We have re-edited the English language of our manuscript.

Reviewer 4 Report

Comments and Suggestions for Authors

Liang, Zhang, and their team presented a review topic entitled “An Overview of Nanoparticle-Based Delivery Platforms for Vaccines”. The manuscript mainly focuses on various nanoparticulate systems including lipid nanoparticles and their vaccination uses. 

Overall, the manuscript can be considered for publication in the MDPI’s Journal Vaccine after significant revisions. We recommend a detailed revision addressing the following issues and concerns carefully to reach a broad audience of disciplines before considering a possible publication.

1.    The review is disjointed and needs to be more cohesive. At the current stage, this paper is a hodgepodge of fragments to some extent. For example, Pfizer/BioNTech and Moderna were used many times in the manuscript in lines 35, 67, 135, and 498. The authors can remove duplicated sentences and introduce the main significance of lipid nanoparticles and their commercial names with company names in one sentence in the introduction. 

For example, Pfizer/BioNtech’s Comirnaty (NT162b2) and Moderna's Spikevax (mRNA-1273), are the first mRNA products to receive approval from the United States Food and Drug Administration (FDA) or the European Medicines Agency (EMA).

2.    Many acronyms and abbreviations are also duplicated. These repeated acronyms will increase the word count without scientific significance. Acronyms and abbreviations must be used for the first time appearance in the text.  

For example, Line 51: mRNA (messenger RNA, mRNA) 

Lines 75, 119, 286, 415, 467: dendritic cell (DC) 

3.    Remove acronyms and abbreviations, if they are used only once in the manuscript. 

For example, Line 377: poly (amino ester) (PACE) 

Lines 350 and 351: poly(amidoamine)s (PAMAM)

Line 336: (TNF-α)) 

4.    The rationale behind delivery carriers (either lipid nanoparticle, polyplex, polyplex micelle, or polymeric carrier) for the effective introduction of mRNA into cells is missing. For example, mRNA is highly susceptible to exo- and endo-ribonuclease-mediated degradation in extracellular and intracellular compartments. To prevent ribonuclease attacks and and promote endo(lyso)somal escape, mRNA has been packaged into the nanoparticles as a core or adsorbed onto the lipid bilayers (J Drug Target 2019;27(5-6):670-680).  indeed, packaging mRNA inside the lipid nanoparticles and polyplex micelles dramatically improved the protection of mRNA by 10000-fold compared to naked mRNA without packaging.

5.    What are polyplexes? How they augment the mRNA protection must be discussed. 

6.    We recommend the authors introduce draw a figure on how mRNA delivery induces vaccinization. This will help the readers quickly understand the story.  The authors provided only one figure with different types of nanoparticles.  We recommend the authors how these nanoparticles help or augment the induction of vaccine effect or immunotherapeutic effects or modes of action of each nanoparticle must be drawn. 

7.    Line 414: What is the letter “E’ representing in arginine-alanine-leucine E-alanine (RALA)? Is E one letter code for glutamic acid?

8.    What are polyplexes? How they augment the mRNA protection must be discussed. 

9.    We recommend introducing the pros and cons of current FDA-approved LNPs. For example, these LNPs tend to accumulate in the liver even after local direct intramuscular injection. 

10. An excellent review article is forward-looking, meaning that the authors can imagine a future where protein-based nanocarriers will be headed in the next 10-years.

What fundamental issues or challenges still need to be addressed? 

What potential solutions can authors suggest to fellow researchers? 

Author Response

Response to Reviewer 4 Comments

1. Summary

2. Point-by-point response to Comments and Suggestions for Authors

Comments 1: The review is disjointed and needs to be more cohesive. At the current stage, this paper is a hodgepodge of fragments to some extent. For example, Pfizer/BioNTech and Moderna were used many times in the manuscript in lines 35, 67, 135, and 498. The authors can remove duplicated sentences and introduce the main significance of lipid nanoparticles and their commercial names with company names in one sentence in the introduction.

For example, Pfizer/BioNtech’s Comirnaty (NT162b2) and Moderna's Spikevax (mRNA-1273), are the first mRNA products to receive approval from the United States Food and Drug Administration (FDA) or the European Medicines Agency (EMA).

Response 1: Thank you for bringing this to our attention. The expression approach has been revised and repetitive expressions have been eliminated in the line 186-191.

Comments 2: Many acronyms and abbreviations are also duplicated. These repeated acronyms will increase the word count without scientific significance. Acronyms and abbreviations must be used for the first appearance in the text.

For example, Line 51: mRNA (messenger RNA, mRNA).

Lines 75, 119, 286, 415, 467: dendritic cell (DC).

Response 2: Thank you for pointing this out. The incorrect writing on our part is deeply regretted, I have changed the abbreviation (DC) for dendritic cells in line 74.

Comments 3: Remove acronyms and abbreviations, if they are used only once in the manuscript.

For example, Line 377: poly (amino ester) (PACE).

Lines 350 and 351: poly(amidoamine)s (PAMAM).

Line 336: (TNF-α)).

Response 3: It is really a careless mistake to the whole quality of our article. We have corrected them and we also feel great thanks for your point out.

Comments 4: The rationale behind delivery carriers (either lipid nanoparticle, polyplex, polyplex micelle, or polymeric carrier) for the effective introduction of mRNA into cells is missing. For example, mRNA is highly susceptible to exo- and endo-ribonuclease-mediated degradation in extracellular and intracellular compartments. To prevent ribonuclease attacks and and promote endo(lyso)somal escape, mRNA has been packaged into the nanoparticles as a core or adsorbed onto the lipid bilayers (J Drug Target 2019;27(5-6):670-680).  indeed, packaging mRNA inside the lipid nanoparticles and polyplex micelles dramatically improved the protection of mRNA by 10000-fold compared to naked mRNA without packaging.

Response 4: Thanks for your suggestions, we have added various delivery platforms how to effectively deliver mRNA into cells, we hope it meets your requirements.

Comments 5: What are polyplexes? How they augment the mRNA protection must be discussed.

Response 5: I would like to sincerely apologize for any confusion that may have arisen due to the misinterpretation of my statement. The cationic polymer can compress mRNA into nanoparticles with high density to obtain stable polyplexes. And I have made relevant explanations in the revised article in the line 407-413.

Comments 6: We recommend the authors introduce draw a figure on how mRNA delivery induces vaccinization. This will help the readers quickly understand the story.  The authors provided only one figure with different types of nanoparticles.  We recommend the authors how these nanoparticles help or augment the induction of vaccine effect or immunotherapeutic effects or modes of action of each nanoparticle must be drawn.

Response 6: Thank you for your recommendation. In order to be more clearly about the different types of nanoparticles, we have added the chemical formulas in Figure 2.

Comments 7: Line 414: What is the letter “E’ representing in arginine-alanine-leucine E-alanine (RALA)? Is E one letter code for glutamic acid?

Response 7: I apologize for my lack of attention. I have rectified the spelling error in line 489.

Comments 8: What are polyplexes? How they augment the mRNA protection must be discussed.

Response 8: I would like to sincerely apologize for any confusion that may have arisen due to the misinterpretation of my statement. The cationic polymer can compress mRNA into nanoparticles with high density to obtain stable polyplexes. And I have made relevant explanations in the revised article in the line 404-421.

Comments 9: We recommend introducing the pros and cons of current FDA-approved LNPs. For example, these LNPs tend to accumulate in the liver even after local direct intramuscular injection.

Response 9: Thanks for your great suggestion on improving the accessibility of our manuscript. We have added about the pros and cons of current FDA-approved LNPs in the line 314-324. We hope you will find this revised version satisfactory.

Comments 10: An excellent review article is forward-looking, meaning that the authors can imagine a future where protein-based nanocarriers will be headed in the next 10-years.

What fundamental issues or challenges still need to be addressed?

What potential solutions can authors suggest to fellow researchers?

Response 10: Thanks for your suggestions, we have added protein-based delivery platforms about their future in the line 503-508, we hope it meets your requirements.

Round 2

Reviewer 2 Report

Comments and Suggestions for Authors

The authors took into account the reviewer’s suggestions

Author Response

We sincerely thank you for previous thoroughly examining our manuscript and providing very helpful comments to guide our revision. We appreciated for reviewers’ warm work earnestly, and hope that the connection will meet with approval. Once again, thank you very much for your comments and suggestions.

Reviewer 3 Report

Comments and Suggestions for Authors

Some previous comments were not addressed:

1. Include in this section how are the LNPs stabilized, electrostatically or sterically?

2. Section 3.2, please describe how the delivery systems are prepared. How they are modified for targeting. How they are stabilized. The role of PEG?

3. Section 3.3, please describe how the delivery systems are prepared. How they are modified for targeting. How they are stabilized. The role of PEG?

The authors are clearly avoiding the main topic associated to nanoparticles: nanoparticles stability, which implies that they have no idea in this regard.

Line 208, "the surface of LNPs is easily modifiable". This is not completely true. The answer depends on if after running pre-modification or post-modification the nanosystems are stable or not? The problem is that the authors need to address this issue. A section dealing with nanoparticle stability must be added. In there, talk about steric and electrostatic stabilization. The role of PEG. If post-modification is pursued, what happens with the stability of the modified nanoparticle? What do they recommend? Stabilize during nanoparticle preparation with pre-modified materials? Or afterward? What happens to particle stability if targeting is pursued? Should we do the targeting pre- or post-modification? Should we pursued targeting and stabilization at once?

Nanoparticle stability cannot be removed from a nanoentity, this concept is continually omitted in the literature. If a nanoparticle (nanovaccine or whatever) is not stable, it is useless. 

Comments on the Quality of English Language

The English language significantly improved, there are still minor errors. 

Author Response

For review article

Response to Reviewer 3 Comments

1. Summary

We sincerely thank you for thoroughly examining our manuscript and providing very helpful comments to guide our revision. We have tried our best to revise the manuscript according to your kind and constructive comments and suggestions. To be more clearly and in accordance with the reviewer concerns, we have added some details about stability and modification of lipid-based nanoparticle on section 3.1.2. We sincerely hope that this revised manuscript has addressed all your comments and suggestions. We appreciated for reviewers’ warm work earnestly, and hope that the connection will meet with approval. Once again, thank you very much for your comments and suggestions.

2. Point-by-point response to Comments and Suggestions for Authors

Comments 1: Include in this section how are the LNPs stabilized, electrostatically or sterically?

Response 1: Thank you for your recommendation. We have discussed these contents in section 3.1.2 (Stability and modification of lipid-based nanoparticle delivery platform) in accordance with your suggestion.

Comments 2: Section 3.2, please describe how the delivery systems are prepared. How they are modified for targeting. How they are stabilized. The role of PEG?

Response 2: We eagerly anticipate receiving your insightful comments. The inclusion of preparation methods and the elucidation of PEG's role have been incorporated into this section in line 438-460.

Comments 3: Section 3.3, please describe how the delivery systems are prepared. How they are modified for targeting. How they are stabilized. The role of PEG?

Response 3: We sincerely appreciate the valuable comments. We have added some details in the line 512-531.

3. Response to Comments on the Quality of English Language

Point 1: A native English speaker must review this manuscript.

Response 1: Thank you for thoroughly examining our manuscript. We have re-edited the English language of our manuscript.

Reviewer 4 Report

Comments and Suggestions for Authors

The authors satisfactorily addressed all the comments raised by the reviewer.  Hence, the manuscript is accepted for publication. 

Author Response

(The authors gave the same response as above.)

Round 3

Reviewer 3 Report

Comments and Suggestions for Authors

The authors have addressed all my previous concerns. 

Comments on the Quality of English Language

Moderate review of the English language is required.